# Mortality in rheumatoid arthritis patients with pulmonary nontuberculous mycobacterial disease: A retrospective cohort study

**Shunsuke Mori**[1]*, **Yukinori Koga**[2], **Kazuyoshi Nakamura**[3], **Sayuri Hirooka**[3], **Takako Matsuoka**[3], **Hideshi Uramoto**[3], **Osamu Sakamoto**[3], **Yukitaka Ueki**[4]

1 Department of Rheumatology, Clinical Research Center for Rheumatic Diseases, National Hospital Organization Kumamoto Saishun Medical Center, Kohshi, Kumamoto, Japan, 2 Department of Radiology, Clinical Research Center for Rheumatic Diseases, National Hospital Organization Kumamoto Saishun Medical Center, Kohshi, Kumamoto, Japan, 3 Department of Respiratory Medicine, National Hospital Organization Kumamoto Saishun Medical Center, Kohshi, Kumamoto, Japan, 4 Rheumatic and Collagen Disease Center, Sasebo Chuo Hospital, Sasebo, Nagasaki, Japan

* mori.shunsuke.ra@mail.hosp.go.jp

**Data Availability Statement:** All data underlying the findings are available from the Human Research Ethics Committee of the National

## Abstract

### Objective

The aim of this study was to compare long-term mortality following diagnosis of pulmonary nontuberculous mycobacterial (NTM) disease between patients with and without rheumatoid arthritis (RA) and to evaluate predictive factors for death outcomes.

### Methods

We reviewed the electronic medical records of all patients who were newly diagnosed with pulmonary NTM disease at participating institutions between August 2009 and December 2018. Patients were followed until death, loss to follow-up, or the end of the study. Taking into consideration the presence of competing risks, we used the cumulative incidence function with Gray's test and Fine-Gray regression analysis for survival analysis.

### Results

A total of 225 patients (34 RA patients and 191 non-RA controls) were followed, with a mean time of 47.5 months. Death occurred in 35.3% of RA patients and 25.7% of non-RA patients. An exacerbation of pulmonary NTM disease represented the major cause of death. The estimated cumulative incidence of all-cause death at 5 years was 24% for RA patients and 23% for non-RA patients. For NTM-related death, the 5-year cumulative incidence rate was estimated to be 11% for RA patients and 18% for non-RA patients. Gray's test revealed that long-term mortality estimates were not significantly different between patient groups. Fine-Gray regression analysis showed that the predictive factors for NTM-related death were advanced age (adjusted hazards ratio 7.28 [95% confidence interval 2.91–18.20] for ≥80 years and 3.68 [1.46–9.26] for 70–80 years vs. <70 years), male sex (2.40 [1.29–4.45]), *Mycobacterium abscessus* complex (4.30 [1.46–12.69] vs. *M. avium*), and cavitary disease (4.08 [1.70–9.80]).

Hospital Organization Kumamoto Saishun Medical Center for all interested researchers who meet the criteria for access to confidential data. Since these data include potentially identifying or sensitive personal information of individual patients, however, the Committee does not recommend that such data be made public unnecessarily. Please contact Mr. Shunichi Tsutsumiuchi, the Control Manager of the Committee, at tsutsumiuchi. shunichi.dz@mail.hosp.go.jp to request the data.

**Funding:** The study was supported by research funds from the National Hospital Organization, Japan. The funder had no role in study design, data collection and analysis, decision to publish, or preparation of the manuscript.

**Competing interests:** The authors have declared that no competing interests exist.

## Conclusions

RA patients with pulmonary NTM disease were not at greater risk of long-term mortality compared with non-RA patients. Rather, advanced age, male sex, causative NTM species, and cavitary NTM disease should be considered when predicting the outcomes of RA patients with pulmonary NTM disease.

## Introduction

Nontuberculous mycobacteria (NTM) are typically opportunistic pathogens that are ubiquitous in natural and man-made environments. While NTM species can cause a wide variety of skin, soft-tissue, and osteoarticular infections as well as superficial lymphadenitis, its most common infection site is the lung [1]. Precise data on the prevalence or incidence of pulmonary NTM disease are limited by the fact that, unlike tuberculosis (TB), NTM disease reporting is not mandatory. Nevertheless, there has been increasing awareness among clinicians that pulmonary NTM disease is becoming more prevalent and can be seen as an emerging public health problem [2, 3]. According to nationwide surveys conducted in industrialized countries, the incidence of TB has decreased or stabilized, but the annual prevalence or incidence rate of pulmonary NTM disease is increasing and now exceeds that of TB [4–9]. In addition, population-based studies conducted in Japan and the United States have reported that the number of NTM-related deaths is increasing, although there is significant geographical variation in each country [10, 11]. In a recent population-based comparative study in Korea, NTM-infected patients had poor prognosis compared with TB patients or the general population [12].

The increased risk of pulmonary NTM disease in patients with rheumatoid arthritis (RA) has been reported worldwide. Compared with non-RA patients, the adjusted hazard ratios (HRs) for RA patients were calculated as 2.07 in Canada and 4.17 to 6.24 in Taiwan [13–15]. A high prevalence of pulmonary complications, such as interstitial lung disease (ILD) and airway disease, is well known in RA patients [16–18], and the risk for developing TB is also higher among RA patients than among the general population [14, 15]. Structural abnormalities in the lungs associated with these conditions are considered one of the major host risk factors for pulmonary NTM disease [19]. In addition, reports of NTM disease are emerging in RA patients undergoing treatment with anti-tumor necrosis factor (anti-TNF) agents [20–23]. In the United States, the crude incidence rate (100,000 person-years [PYs]) of pulmonary NTM disease is significantly higher in the anti-TNF-exposed RA population compared with unexposed RA patients (105 vs. 19.2) and the general population (105 vs. 4.1) [21]. Oral prednisolone was also reported as a risk factor for developing NTM disease in RA patients [19, 23]. It is evident that data on incidence rates and risk factors for the development of pulmonary NTM disease in the RA population is accumulating. Nevertheless, it remains unknown if RA may increase the risk of mortality in patients who have developed pulmonary NTM disease or whether RA-specific factors may be associated with mortality in patients with pulmonary NTM disease.

To address these issues, we performed a retrospective cohort study on RA and non-RA patients who were newly diagnosed with pulmonary NTM disease in the divisions of rheumatology and respiratory medicine of our institutions between August 2009 and December 2018. Taking into consideration the presence of competing risks, we used the cumulative incidence function (CIF) and Gray's test for survival analysis. We estimated

the cumulative incidence rates of all-cause death and NTM-related death over time and compared these mortality estimates between RA and non-RA patients. To evaluate the effect of baseline patient characteristics on death outcome and to calculate an adjusted HR for each predictor variable, we performed Fine-Gray competing risks regression analysis.

## Materials and methods

### Patients

The present study included all RA patients who were newly diagnosed with pulmonary NTM disease in the rheumatology divisions of the following community hospitals in Japan between August 2009 and December 2018: National Hospital Organization (NHO) Kumamoto Saishun Medical Center and Sasebo Chuo Hospital. All patients were required to fulfill the 1987 American College of Rheumatology (ACR) criteria or the 2010 ACR/European League Against Rheumatism (EULAR) criteria for diagnosis of RA [24, 25]. We defined pulmonary NTM disease according to both the 2008 diagnostic criteria of pulmonary NTM disease proposed by the Japanese Society for Tuberculosis (JST) and the Japanese Respiratory Society (JRS) and the 2007 diagnostic criteria for NTM lung disease proposed by the American Thoracic Society (ATS) and the Infectious Disease Society of America (IDSA); namely, patients must have positive culture results from two or more sputum samples obtained separately (or at least one isolation of NTM in the case of bronchoscopy specimens) plus one or more of the following radiological findings: nodular opacities, dissemination of small nodular or branching opacities, homogeneous opacities, cavitary lesions, and bronchiectasis or bronchiolectasis. In addition, other diagnoses must be excluded [26, 27]. As non-RA controls, we registered all patients without RA who were newly diagnosed with pulmonary NTM disease in the respiratory disease division of NHO Kumamoto Saishun Medical Center during the same period. Among non-RA patients in our cohort, only one had a diagnosis of autoimmune rheumatic disease (ANCA-associated vasculitis). Participants in this study were required to be 18 years of age or over.

### Study design

We reviewed patients' electronic medical records to scrutinize clinical data at baseline and during follow-up periods. For each patient, baseline data were obtained at the time when the diagnosis of pulmonary NTM disease was determined, which included demographic characteristics, RA-related data (disease duration, radiological Steinbrocker's stages, and treatment regimens for RA), comorbidities (type 2 diabetes, malignancy, and ILD), TB history, and laboratory data (serum albumin levels and lymphocyte count). Patient hypoalbuminemia and lymphocytopenia were graded according to the National Cancer Institute's Common Terminology Criteria for Adverse Events (NCI-CTCAE) version 4.0.

Causative NTM isolates were determined at the same time. RA therapies that patients were receiving when pulmonary NTM was first suspected were deemed before-diagnosis therapies and those that patients decided to continue or start at the time of the diagnosis of pulmonary NTM were deemed after-diagnosis therapies. Follow-up started on the day of pulmonary NTM disease diagnosis and ended with death, loss to follow-up, or the last follow-up visit before February 1, 2020, whichever was first. Patients who missed two or more scheduled visits without any contact were classified as lost to follow-up.

## Determination of HRCT patterns of pulmonary NTM disease

For radiological classification of pulmonary NTM disease, we reviewed high-resolution computed tomography (HRCT) images taken when the diagnosis of pulmonary NTM disease was determined. HRCT abnormalities included nodules (centrilobular small nodules and nodules greater than 10 mm in diameter), tree-in-bud sign, airspace consolidation, bronchiectasis/bronchiolectasis, and cavities [28]. Based on the predominant HRCT findings and their distributions, each patient was classified as having one of the following three forms: nodular bronchiectasis (NB) form, fibrocavitary form, and unclassifiable form. The NB form was defined by the presence of multifocal bronchiectasis and clusters of small nodules, which was further classified into cavitary NB and non-cavitary NB according to the presence of cavitary lesions on HRCT. The fibrocavitary form was defined by the presence of cavitary lesions often associated with pleural thickening predominantly in the upper lobes [29]. In this study, cavitary NB form and fibrocavitary form were combined and statistically analyzed as the cavitary disease group because (1) it has been reported that both forms have a similar prognosis [29–31]; (2) we focused on the influence of presence of cavitary lesions on mortality estimates over time; and (3) the number of patients with the fibrocavitary form was small. If the disease did not fall under either the NB form or the fibrocavitary form, it was considered unclassifiable.

## Anti-NTM therapy

The following strategy was routinely used as anti-NTM therapy in the participating division. For patients with pulmonary *Mycobacterium avium* complex (MAC) disease, anti-NTM treatment based on triple therapy was started with clarithromycin (600–800 mg/day), ethambutol (15 mg/kg/day, maximum 750 mg), and rifampicin (10 mg/kg/day, maximum 600 mg) according to the 2012 JST/JRS guidelines for anti-NTM chemotherapy. If necessary, streptomycin or kanamycin was added [32]. For patients with NTM disease caused by other NTM species, treatment regimens were decided by the treating physician. The timing of treatment commencement was decided on a case-by-case basis after due consideration of the risk-and-benefit balance of the treatment. The primary endpoint of treatment was set as 12 months of negative sputum cultures while on therapy [26, 32]. The number of patients who had continued anti-NTM therapy until culture-negative sputum had been maintained for 12 months (completion of anti-NTM therapy) was recorded during follow-up.

For RA patients who were given the diagnosis of pulmonary NTM disease but decided to continue or start RA therapy with a biological disease-modifying antirheumatic drug (bDMARD) or a targeted synthetic DMARD (tsDMARD), anti-NTM drugs were used concomitantly during follow-up under strict physician supervision with full patient compliance.

## Cause of death

Primary cause of death, which was defined as the disease or event that started the chain of events that led to death, was determined according to each treating physician's judgment. NTM-related death was defined as death caused by an exacerbation of pulmonary NTM disease based on radiological findings. Final diagnosis was determined by consensus.

## Ethical approval

This study was conducted in accordance with the principles of the Declaration of Helsinki (2008). The protocol of this study also meets the requirements of the Ethical Guidelines for Medical and Health Research Involving Human Subjects, Japan (2014), and has been

approved by the Human Research Ethics Committee of the NHO Kumamoto Saishun Medical Center (No. 29-45/29-45-2) and the Institutional Review Board of Sasebo Chuo Hospital (No. 2014–12). Since the study involved a retrospective review of patient records and the data were analyzed anonymously, our ethical committees waived the requirement of patient informed consent to participate.

## Statistical analysis

To compare baseline patient characteristics between the RA and non-RA groups, we performed the chi-square test or Fisher's exact probability test for categorial variables and by the independent-measures *t*-test for continuous variables. Crude incidence rates of death and 95% confidence intervals (CIs) were calculated by dividing the number of incidence cases by the number of corresponding follow-up PYs overall for both patient groups.

Cumulative incidence of death is defined as the probability that a death event has occurred before a given time. In the present study, we used the CIF to estimate the provability of the occurrence of a death event over time, because we considered the presence of competing risks (namely, lost to follow-up in the survival analysis of all-cause death outcome; lost to follow-up and other-cause death in the survival analysis of NTM-related death outcome). The occurrence of a competing risk event precludes the occurrence of the primary event of interest. In the absence of competing risks, the Kaplan-Meier survival function can be used to estimate the probability of death over time. In the presence of competing risks, however, the simple use of the Kaplan-Meier survival function can overestimate the cumulative incidence probability of all-cause and NTM-related death. To avoid this possibility, we used the CIF instead of the Kaplan-Meier survival function. Gray's test for the CIF model, with or without the *post hoc* Holm's procedure, was used to test the equality of CIF plots among two or three patient groups. Gray's test is the analogue to the log-rank test that is used for testing the equality of Kaplan-Meier survival curves between groups [33, 34].

Fine-Gray competing risks regression analysis was used to evaluate the effect of each of the baseline patient characteristics on all-cause death and NTM-related death over time and to calculate adjusted HRs with 95% CIs. To reduce the possibility that variables with the clinical relevance and importance might be missed out, we first screened all predictive variables with Gray's test as univariate analysis. Predictor variables with *p*-values <0.1 in Gray's test were then employed in a Fine-Gray regression analysis. To compensate for the small number of death events, a backward stepwise selection with a cut-off significance level of 0.05 was used as the variable selection procedure in the Fine-Gray analysis. The proportional hazards assumption was checked using log-minus-log plots of the log cumulative hazard curve function and scaled Schoenfeld residual plots for exposure variables over time. By calculating the variance inflation factor, we also examined if there was any multicollinearity among predictor variables.

For all tests, probability values (*p* values) <0.05 were considered to indicate statistical significance. All calculations were performed using PASW Statistics version 22 (SPSS Japan Inc., Tokyo, Japan) and Easy R (Saitama Medical Center, Jichi Medical University, Saitama Japan) [35].

## Results

### Baseline characteristics of patients who were newly given a diagnosis of pulmonary NTM disease

A total of 225 patients were newly diagnosed with pulmonary NTM disease at participating institutions between August 2009 and December 2018. Baseline characteristics are shown in

Table 1. Among them, there were 34 RA patients and 191 non-RA controls. In RA patients, the mean duration of RA was 11.6 years (95% CI 7.7–15.5), and approximately two-thirds were at Steinbrocker's radiological stages III and IV. All RA patients were receiving pharmaceutical RA therapy; among them, 11 patients (32.4%) and one patient (2.9%) were being treated with a biological DMARD (infliximab, etanercept, adalimumab, abatacept, or tocilizumab) and a tsDMARD (tofacitinib), respectively. Nine RA patients continued or started pharmaceutical therapies with a bDMARD or tofacitinib, together with anti-NTM drugs, after the diagnosis of pulmonary NTM disease was made. Females were predominant in the RA group compared with the control group (88.2% vs. 67.5%). There were no significant differences in rates of comorbidities or past TB. The mean lymphocyte count was lower in RA patients compared with non-RA patients. The most common NTM isolate was *M. intracellulare*; it was isolated in 50% of RA patients and 67.0% of non-RA patients. *M. avium* was isolated in 38.2% of RA patients and 24.1% of non-RA patients, and the *M. abscessus* complex was isolated in 11.8% of RA patients and 5.5% of non-RA patients. In both patient groups, the most predominant HRCT pattern was the non-cavitary NB form (44.1% of RA patients and 42.9% of non-RA patients), followed by the cavitary NB form (29.4% and 28.3%), the unclassifiable form (23.5% and 15.2%), and the fibrocavitary form (2.9% and 13.6%). Cavitary lesions also existed at similar rates on HRCT between the RA and non-RA groups (32.4% vs. 41.9%). There were no significant differences in the rates of other abnormal findings on HRCT scans, such as the tree-in-bud sign, consolidation, nodules, or bronchiectasis/bronchiolectasis. After the diagnosis of pulmonary NTM disease, approximately 45% of patients in both groups completed anti-NTM therapy.

## Mortality in RA and non-RA patients with pulmonary NTM disease

After the diagnosis of pulmonary NTM disease was newly made, patients were followed for a mean time of 47.5 months (95% CI 42.9–52.0). Sixty-nine patients (2 RA patients and 67 non-RA patients) were lost during follow-up. As shown in Table 2, death occurred in a total of 61 patients (12 cases in the RA group and 49 cases in the non-RA group). An exacerbation of pulmonary NTM disease represented the major cause of death in the RA and non-RA patient groups (50% vs. 73.5%, respectively). In RA patients, respiratory failure due to an exacerbation of ILD accounted for one-quarter of the causes of death.

As shown in Table 3, the crude incidence rate of all-cause death was 6.9 per 100 PYs (95% CI 3.9–12.1) for RA patients and 6.9 per 100 PYs (95% CI 5.2–9.1) for non-RA patients. For

**Table 1. Baseline characteristics of patients who were newly diagnosed with pulmonary NTM disease (n = 225).**

|  | RA patients (n = 34) | Non-RA patients (n = 191) | $p^*$ |
|---|---|---|---|
| Age, years, mean (95% CI) | 70.6 (67.1–74.2) | 70.7 (69.1–72.3) | 0.98 |
| ≥80, number (%) | 8 (23.5) | 46 (24.1) | 1.00 |
| ≥70 and <80, number (%) | 12 (35.3) | 57 (29.8) | 0.55 |
| <70, number (%) | 14 (41.2) | 88 (46.1) | 0.71 |
| Male, number (%) | 4 (11.8) | 62 (32.5) | 0.014 |
| RA duration, years, mean (95% CI) | 11.6 (7.7–15.5) | – | – |
| Steinbrocker's stages III/IV, number (%) | 23 (67.6) | – | – |
| RA therapies[†], before/after diagnosis, number (%) | 34 (100) / 34 (100) | – | – |
| MTX therapy | 10 (29.4) / 8 (23.5) | – | – |
| csDMARD (except MTX) therapy | 5 (14.7) / 8 (23.5) | – | – |
| bDMARD therapy (with or without MTX) | 11 (32.4) / 8 (23.5) | – | – |
| tsDMARD therapy (with or without MTX) | 1 (2.9) / 1 (2.9) | – | – |

*(Continued)*

**Table 1.** (Continued)

| | RA patients (n = 34) | Non-RA patients (n = 191) | $p^*$ |
|---|---|---|---|
| No DMARD use (oral steroids and/or NSAIDs) | 7 (20.6) / 9 (26.5) | – | – |
| Comorbidity, number (%) | | | |
| Type 2 diabetes | 4 (11.8) | 24 (12.6) | 1.00 |
| Malignancy | 1 (2.9) | 26 (13.6) | 0.090 |
| Interstitial lung disease | 4 (11.8) | 15 (7.9) | 0.50 |
| Tuberculosis history, number (%) | 1 (2.9) | 9 (4.7) | 1.00 |
| Serum albumin, g/dl, mean (95% CI) | 3.6 (3.4–3.8) | 3.8 (3.7–3.8) | 0.19 |
| <3.0 | 4 (11.8) | 20 (10.5) | 0.77 |
| ≥3.0 and <4.0 | 21 (61.8) | 88 (46.1) | 0.10 |
| ≥4.0 | 9 (26.5) | 83 (43.5) | 0.09 |
| Lymphocyte count, /mm$^3$, mean (95% CI) | 1195 (1047–1343) | 1394 (1320–1468) | 0.036 |
| <800 | 7 (20.6) | 21 (11.0) | 0.15 |
| ≥800 and <1000 | 5 (14.7) | 20 (10.5) | 0.55 |
| ≥1000 | 22 (64.7) | 150 (78.5) | 0.12 |
| Causative NTM species, number (%) | | | |
| *M. avium* | 13 (38.2) | 46 (24.1) | 0.093 |
| *M. intracellulare* | 17 (50) | 128 (67.0) | 0.079 |
| *M. abscessus* complex | 4 (11.8) | 11 (5.8) | 0.25 |
| Other species[‡] | 0 | 6 (3.1) | 0.60 |
| HRCT patterns of NTM disease, number (%) | | | |
| Cavitary NB form | 10 (29.4) | 54 (28.3) | 1.00 |
| Non-cavitary NB form | 15 (44.1) | 82 (42.9) | 1.00 |
| Fibrocavitary form | 1 (2.9) | 26 (13.6) | 0.090 |
| Unclassifiable form | 8 (23.5) | 29 (15.2) | 0.22 |
| Presence of abnormal HRCT findings, number (%) | | | |
| Cavitary lesion | 11 (32.4) | 80 (41.9) | 0.35 |
| Tree-in-bud sign | 18 (53.0) | 110 (57.6) | 0.71 |
| Consolidation | 15 (44.1) | 77 (40.3) | 0.71 |
| Nodules | 32 (94.1) | 179 (93.7) | 1.00 |
| Bronchiectasis/bronchiolectasis | 25 (73.5) | 159 (83.2) | 0.23 |
| Completion rate of anti-NTM therapy[§], number (%) | 15 (44.1) | 87 (45.5) | 1.00 |

Data were obtained at the time of pulmonary NTM disease diagnosis.

*Comparisons of baseline characteristics between the RA and non-RA groups were performed using the chi-square test or Fisher's exact probability test for categorial variables and by the independent-measures *t*-test for continuous variables.

[†]RA therapies represent those that patients were receiving when pulmonary NTM was first suspected (before diagnosis) and those that patients decided to continue or restart at the time of pulmonary NTM disease diagnosis (after diagnosis). bDMARDs included etanercept, infliximab, adalimumab, abatacept, and tocilizumab. csDMARDs included tacrolimus and salazosulfapyridine. tsDMARD was tofacitinib.

[‡]Other species included *M. gordonae* (n = 3), *M. fortuitum* (n = 2), and *M. szulgai* (n = 1).

[§]Treatment completion rate was defined as the number (%) of participants who had continued anti-NTM therapy until the primary treatment endpoint (culture-negative sputum for 12 months). This value was determined during follow-up.

RA, rheumatoid arthritis; MTX, methotrexate; DMARD, disease-modifying antirheumatic drug; csDMARD, conventional synthetic DMARD; bDMARD, biological DMARD; tsDMARD, targeted synthetic DMARD; NSAIDs, non-steroidal anti-inflammatory drugs; NTM, nontuberculous mycobacteria; NB form, nodular/bronchiectatic form; HRCT, high-resolution computed tomography.

**Table 2. Cause of death in patients with pulmonary NTM disease (n = 61).**

| | RA patients (n = 12) | Non-RA patients (n = 49) |
|---|---|---|
| Causes, number (%) | | |
| Exacerbation of pulmonary NTM disease | 6 (50) | 36 (73.5) |
| Exacerbation of interstitial lung disease | 3 (25) | 3 (6.1) |
| Malignancy | 0 | 5 (10.2) |
| Ischemic heart failure | 1 (8.3) | 0 |
| Pyelonephritis | 1 (8.3) | 0 |
| Cerebral infarction | 0 | 1 (2.0) |
| Subarachnoid hemorrhage | 0 | 1 (2.0) |
| HCV-related liver cirrhosis | 0 | 1 (2.0) |
| Gastrointestinal amyloidosis | 0 | 1 (2.0) |
| Intestinal infectious disease | 0 | 1 (2.0) |
| Myelodysplastic syndrome | 1 (8.3) | 0 |

NTM, nontuberculous mycobacteria; RA rheumatoid arthritis; HCV hepatitis C virus.

NTM-related death, which was defined as death caused by the exacerbation of pulmonary NTM disease, the crude incidence rate was 3.4 per 100 PYs (95% CI 1.5–7.6) for RA patients and 5.0 per 100 PYs (95% CI 3.6–7.0) for non-RA patients.

According to the CIF, which was based on a competing risks model, the overall 5-year cumulative death probability was estimated to be 24% for RA patients and 23% for non-RA patients: the cumulative incidence of all-cause death at 5 years was 0.24 (95% CI 0.10–0.41) for RA patients and 0.23 (95% CI 0.17–0.29) for non-RA patients (Table 3). There were no significant differences in mortality estimates over time between patient groups ($p = 0.36$ with Gray's test). Similarly, the cumulative incidence of NTM-related death at 5 years was not significantly different between RA patients and non-RA patients (0.11 [95% CI 0.03–0.29] vs. 0.18 [95% CI 0.12–0.24], $p = 0.77$). CIF plots for NTM-related death and all-cause death in RA and non-RA patients are shown in Fig 1.

**Table 3. Mortality in patients with pulmonary NTM disease.**

| | All patients (n = 225) | RA patients (n = 34) | Non-RA patients (n = 191) |
|---|---|---|---|
| Follow-up[*], months, mean (95% CI) | 47.5 (42.9–52.0) | 61.7 (67.1–74.2) | 44.9 (40.0–49.8) |
| Lost to follow-up, number (%) | 69 (30.7) | 2 (5.9) | 67 (35.1) |
| All-cause death, number (%) | 61 (27.1) | 12 (35.3) | 49 (25.7) |
| Crude incidence rate per 100 PYs (95% CI) | 6.9 (5.3–8.8) | 6.9 (3.9–12.1) | 6.9 (5.2–9.1) |
| Cumulative incidence at 5 years (95% CI) [†] | 0.22 (0.17–0.28) | 0.24 (0.10–0.41) | 0.23 (0.17–0.29) |
| NTM-related death[‡], number (%) | 42 (18.7) | 6 (17.6) | 36 (18.8) |
| Crude incidence rate per 100 PYs (95% CI) | 5.1 (3.8–6.8) | 3.4 (1.5–7.6) | 5.0 (3.6–7.0) |
| Cumulative incidence at 5 years (95% CI) [†] | 0.16 (0.11–0.22) | 0.11 (0.03–0.29) | 0.18 (0.12–0.24) |

[*] Follow-up was measured from the diagnosis of pulmonary NTM disease.

[†] Cumulative incidences of all-cause death and NTM-related death at 5 years (5-year mortality rates) were estimated by the CIF. Gray's test was used for comparisons of mortality estimates over time between RA patients and non-RA patients ($p = 0.36$ for all-cause death and $p = 0.77$ for NTM-related death).

[‡] NTM-related death was defined as death caused by an exacerbation of the pulmonary NTM disease shown in Table 2.

NTM, nontuberculous, mycobacteria; RA rheumatoid arthritis; PYs, patient-years; CIF, cumulative incidence function; CI, confidence interval.

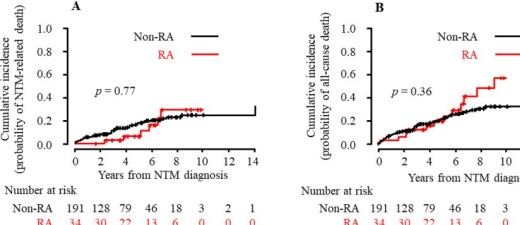

**Fig 1. Cumulative incidence of NTM-related death and all-cause death in RA and non-RA patients.** Using the CIF, the cumulative incidence of NTM-related death (A) and all-cause death (B) in patients who were newly given a diagnosis of pulmonary NTM disease is shown in the RA and non-RA groups. Numbers below these figures represent the number of patients at risk. The cumulative incidence of death over time between both groups was compared using Gray's test. According to univariate Fine-Gray analyses, the unadjusted HR (95% CI) of RA versus non-RA was 0.86 (0.38–1.98, $p$ = 0.73) for NTM-related death and 1.34 (0.75–2.40, $p$ = 0.32) for all-cause death. RA, rheumatoid arthritis; NTM, nontuberculous mycobacterial disease; CIF, cumulative incidence function; HR, hazard ratio; CI, confidence interval.

## Characteristics of death cases in RA patients who developed pulmonary NTM disease during RA treatment

The characteristics of the death cases among RA patients are shown in Table 4. Four cases were receiving a bDMARD when pulmonary NTM was first suspected (before diagnosis; cases 1, 3, 5, and 11). All patients who decided to continue or start RA therapy with biological DMARDs at the start of follow-up concomitantly received anti-NTM therapy (cases 1, 6, and 11). Anti-NTM therapy was completed according to the ATS/IDSA and JST/JRS guidelines in five death cases (cases 1, 3, 5, 7, and 8). During the follow-up period, 55% of patients with advanced age (≥70 years) at baseline, 50% of males, 75% of patients with coexisting ILD, 50% of patients with causative *M. abscessus* complex, and 63.6% of patients with cavitary disease at

**Table 4. Characteristics of RA patients who were newly diagnosed with pulmonary NTM disease and eventually died during follow-up.**

| Case no. | Age/Sex | Causes of death | HRCT patterns of NTM | NTM species | Lung comorbidities‡ | Survival periods (months)* | RA therapies† | |
|---|---|---|---|---|---|---|---|---|
| | | | | | | | **before** | **after** |
| 1 | 80F | NTM exacerbation | Cavitary NB | *M. intracellulare* | – | 80 | MTX/ETN | MTX/ETN |
| 2 | 75F | UIP exacerbation | Unclassifiable | *M. intracellulare* | UIP | 2 | TAC | TAC |
| 3 | 78F | UIP exacerbation | Non-cavitary NB | *M. avium* | UIP | 26 | TCZ | TAC |
| 4 | 70F | NTM exacerbation | Non-cavitary NB | Mab | – | 27 | MTX | MTX |
| 5 | 84F | NTM exacerbation | Cavitary NB | *M. intracellulare* | – | 61 | ETN | TAC |
| 6 | 84F | NTM exacerbation | Cavitary NB | *M. intracellulare* | – | 45 | MTX | ABT |
| 7 | 76M | NTM exacerbation | Cavitary NB | *M. intracellulare* | – | 69 | SASP | SASP |
| 8 | 80F | Heart failure | Unclassifiable | *M. intracellulare* | – | 52 | Steroid | Steroid |
| 9 | 79F | UIP exacerbation | Fibrocavitary | Mab | UIP | 19 | Steroid | Steroid |
| 10 | 81F | Pyelonephritis | Non-cavitary NB | *M. avium* | – | 108 | TAC | TAC |
| 11 | 43M | NTM exacerbation | Cavitary NB | *M. avium* | – | 77 | MTX/ETN | ABT |
| 12 | 80F | Myelodysplasia | Cavitary NB | *M. intracellulare* | – | 92 | MTX | MTX |

*Survival periods represent time intervals between diagnosis of pulmonary NTM disease and death.

†RA therapies represent those that patients were receiving when pulmonary NTM disease was first suspected (before diagnosis) and those that patients decided to continue or start at the time of pulmonary NTM disease diagnosis (after diagnosis).

‡No patients had past TB.

RA, rheumatoid arthritis; NTM nontuberculous mycobacteria; TB, tuberculosis; Mab, *M. abscessus* complex; NB, nodular/bronchiectatic; UIP, unusual interstitial pneumonia; MTX, methotrexate; TAC, tacrolimus; SASP, salazosulfapyridine; ETN, etanercept; TCZ, tocilizumab; ABT, abatacept.

baseline eventually died. Fig 2 shows HRCT scans of case 5 in which the patient died due to an exacerbation of pulmonary NTM disease 61 months after the NTM diagnosis.

### Predictive factors for mortality in patients with pulmonary NTM disease

As shown in Table 3, there was no significant difference in mortality estimates of all-cause death or NTM-related death over time between RA and non-RA patients. We next compared mortality estimates over time between groups of patients classified according to each of the predictor variables. Estimates of cumulative incidence of all-cause death and NTM-related death over time in each group were computed using CIF and compared with Gray's test (Table 5). For all-cause death, advanced age (≥70 years), male sex, ILD, past TB, hypoalbuminemia (<3.0 d/dl), lymphocytopenia (<800/mm$^3$), *M. abscessu*s complex, the presence of cavitary disease (cavitary NB/fibrocavitary form), and abnormal HRCT findings (cavitary lesion and consolidation) were identified as variables significantly associated with mortality estimates over time. For NTM-related death, advanced age (≥70 years), male sex, hypoalbuminemia (<3.0 g/dl), lymphocytopenia (<800/mm$^3$), *M. abscessu*s complex, the presence of cavitary disease (cavitary NB/fibrocavitary form), and abnormal HRCT findings (cavitary lesion) were identified as factors significantly associated with mortality estimates over time. CIF plots grouped according to age, sex, NTM species, and HRCT forms are shown in Fig 3.

All predictor variables with *p* values <0.1 in Gray's test were run through Fine-Gray competing risks regression analysis. The adjusted HRs (95% CI) for the predictor variables of all-cause death and NTM-related death are shown in Table 6. Advanced age (adjusted HR 3.79 [95% CI 1.82–7.89] for ≥80 years and 2.56 [1.27–5.16] for ≥70 and <80 years vs. <70 years), male sex (2.20 [1.27–5.16]), serum albumin <3.0 g/dl (3.16 [1.34–7.44] vs. ≥4.0 g/dl), lymphocyte count <800/mm$^3$ (2.84 [1.41–5.72] vs. 1000/mm$^3$), and cavitary disease (cavitary NB/ fibrocavitary form 2.92 [1.51–5.65] vs. non-cavitary NB form) were identified as significant predictive factors for all-cause death in pulmonary NTM disease patients. For NTM-related death, the predictive factors were advanced age (adjusted HR 7.28 [95% CI 2.91–18.20] for ≥80 years and 3.68 [1.46–9.26] for 70–80 years vs. <70 years), male sex (2.40 [1.29–4.45]), *M. abscessus* complex (4.30 [1.46–12.69] vs. *M. avium*), and cavitary disease (cavitary NB/fibrocavitary form 4.08 [1.70–9.80] vs. non-cavitary NB form). We also performed Fine-Gray competing risks regression analyses for all-cause and NTM-related death using the forced-entry method. Data are shown in S1 Table. We confirmed that there was neither multicollinearity nor violation of the proportional hazards assumption in these predictive factors.

The completion rate of anti-NTM therapy was similar between all-cause death, NTM-related death, and survival cases (45.9%, 40.5%, and 44.2%, respectively), which suggested that this rate might have little effect on all-cause or NTM-related mortality. We were not able to

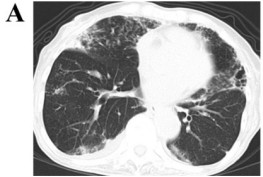
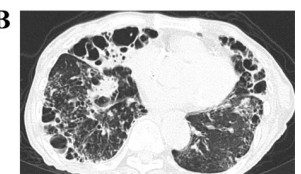

**Fig 2. HRCT scans of a patient with the cavitary NB form (case 5).** (**A**) An HRCT scan taken at the time of diagnosis of pulmonary NTM disease. Nodules and ground-glass opacities are evident in both lungs. Consolidation is evident in the right middle lobe (S4). In addition, bronchiectasis, the tree-in-bud sign, and cavitary lesions are evident in the lingular segment of the left upper lobe (S4). (**B**) An HRCT scan taken 8 months before the patient died. Extensive nodular opacities are evident in both lungs. Bronchiectasis, the tree-in-bud sign, and cavitary lesions are prominent in the right middle lobe, the lingular, and the left lower lobe.

**Table 5. Comparisons of mortality estimates over time between patient groups classified according to each predictor variable.**

| Predictor variables | *p* values | |
|---|---|---|
| | **All-cause death** | **NTM-related death** |
| Age (years) | | |
| ≥80 | <0.001 | <0.001 |
| ≥70 and <80 | 0.001 | 0.002 |
| <70 (reference) | – | – |
| Sex (male vs. female) | 0.002 | 0.003 |
| RA (yes vs. no) | 0.36 | 0.77 |
| Use of bDMARD or tsDMARD | | |
| Prior to diagnosis (yes vs. no) | 0.52 | 0.50 |
| After diagnosis (yes or no) | 0.85 | 0.32 |
| Type 2 diabetes (yes vs. no) | 0.053 | 0.55 |
| Malignancy (yes vs. no) | 0.11 | 0.25 |
| Interstitial lung disease (yes vs. no) | <0.001 | 0.17 |
| Tuberculosis history (yes vs. no) | 0.003 | 0.084 |
| Serum albumin, g/dl | | |
| <3.0 | <0.001 | 0.023 |
| ≥3.0 and <4.0 | 0.08 | 0.11 |
| ≥4.0 (reference) | – | – |
| Lymphocyte count, /mm$^3$ | | |
| <800 | <0.001 | 0.007 |
| ≥800 and <1000 | 0.06 | 0.04 |
| ≥1000 (reference) | – | – |
| Causative NTM species | | |
| *M. abscessus* complex | 0.025 | 0.004 |
| *M. intracellulare* | 0.27 | 0.25 |
| *M. avium* (reference) | – | – |
| HRCT patterns of NTM disease | | |
| Cavitary NB/fibrocavitary form | <0.001 | <0.001 |
| Unclassifiable form | 0.30 | 0.93 |
| Non-cavitary NB form (reference) | – | – |
| Cavitary lesion (yes vs. no) | <0.001 | <0.001 |
| Tree-in-bud sign (yes vs. no) | 0.050 | 0.74 |
| Consolidation (yes vs. no) | 0.018 | 0.091 |
| Nodule (yes vs. no) | 0.071 | 0.84 |
| Bronchiectasis (yes vs. no) | 0.52 | 0.44 |

Mortality estimates (cumulative incidence rates) over time were compared between patient groups using Gray's test for CIF plots. In the case of multiple comparisons, the *post hoc* Holm's procedure was used in the Gray's test. The completion rate of anti-NTM therapy was unable to be compared in these analyses because this variable was determined during follow-up (i.e., it was a time-varying covariate).

RA, rheumatoid arthritis; DMARD, disease-modifying antirheumatic drug; bDMARD, biological DMARD; tsDMARD, targeted synthetic DMARD; NTM nontuberculous mycobacteria; NB form, nodular/bronchiectatic form; CIF, cumulative incidence function.

include this variable in the Fine-Gray analyses because it was determined during follow-up (time-varying covariate).

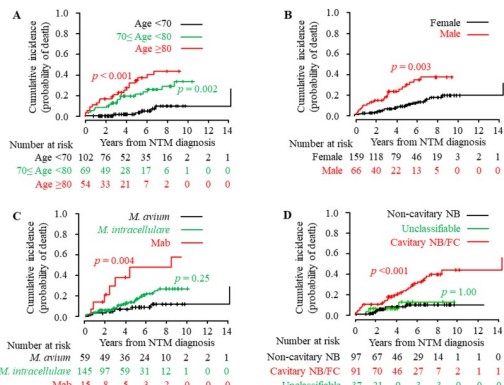

**Fig 3. Cumulative incidence of NTM-related death grouped by predictive factors.** Using the CIF, the cumulative incidence of NTM-related death in patients who were newly diagnosed with pulmonary NTM disease is shown grouped according to predictive factors for death. Predictive factors included (A) age (≥80 years and 70–80 years vs. <70 years), (B) sex (male vs. female), (C) NTM species (*M. abscessu*s complex [Mab] and *M. intracellulare* vs. *M. avium*), and (D) HRCT patterns (cavitary NB/fibrocavitary form and unclassifiable form vs. non-cavitary NB form). Numbers below these figures represent the number of patients at risk. The cumulative incidence of death over time between groups with and without predictive factors was compared using Gray's test with or without the *post hoc* Holm's procedure. NTM, nontuberculous mycobacterial disease; Mab, *M. abscessus* complex; NB, nodular bronchiectatic form; FC form, fibrocavitary form; CIF, cumulative incidence function.

## Discussion

In this retrospective cohort study for patients who were newly diagnosed with pulmonary NTM disease, the crude incidence rate of NTM-related mortality was not greater in the RA group compared with the non-RA group. The estimated cumulative incidence of NTM-related death over time was also similar between groups. Through Fine-Gray competing risks regression analysis, advanced age (≥70 years), male sex, *M. abscessu*s complex, and the presence of cavitary disease were identified as the predictive factors for NTM-related death. Similarly, there was no significant difference in crude incidence rates or longitudinal mortality estimates of all-cause death between RA and non-RA patient groups. The predictive factors for all-cause mortality were advanced age (≥70 years), male sex, hypoalbuminemia, lymphocytopenia, and the presence of cavitary disease.

There remains a dearth of information regarding incidence rates and predictive factors of NTM-related death in RA patients who have developed pulmonary NTM disease. Yamakawa et al. reported that, during a median follow-up of 4.4 years, all-cause death occurred in 38 out of 98 RA patients (38.8%) who were newly diagnosed with pulmonary NTM disease at a single center in Japan between 1993 and 2011, with a cumulative incidence rate of 0.34 at 5 years [31]. They also demonstrated that the presence of cavitary disease (cavitary NB/fibrocavitary form) was a negative prognostic factor for all-cause mortality. Although the incidence rate of NTM-related death was not determined in that study, the results were not inconsistent with our findings. In the present study, 35.3% of RA patients died from any cause during follow-up with a mean of 61.7 months, and when focused on RA patients with cavitary disease at baseline, 63.6% had a death outcome. In a case-control study for RA patients with and without NTM disease who were identified at a single center in Taiwan between 2001 and 2014, Liao et al. showed that eight out of 50 RA patients (16%) died a mean of 1.1 years after NTM infection, and among them, 6 cases had pulmonary NTM disease. Male gender and advanced age were factors associated with mortality [19].

There have been several studies focusing on patients under treatment with bDMARDs. By reviewing data from Northern California Kaiser Permanente, Winthrop et al. identified 18

**Table 6. Predictive factors for mortality in patients with pulmonary NTM disease.**

| Predictor variables | All-cause death | | NTM-related death | |
|---|---|---|---|---|
| | Adjusted HRs (95% CIs) * | *p* | Adjusted HRs (95% CIs) * | *p* |
| Age, years | | | | |
| ≥80 | 3.79 (1.82–7.89) | <0.001 | 7.28 (2.91–18.20) | <0.001 |
| ≥70 and <80 | 2.56 (1.27–5.16) | 0.008 | 3.68 (1.46–9.26) | 0.006 |
| <70 | 1 (reference) | – | 1 (reference) | – |
| Male vs. female | 2.20 (1.21–3.99) | 0.010 | 2.40 (1.29–4.45) | 0.006 |
| Serum albumin, g/dl | | | | |
| <3.0 | 3.16 (1.34–7.44) | 0.009 | – | – |
| ≥3.0 and <4.0 | 1.29 (0.66–2.51) | 0.46 | – | – |
| ≥4.0 (reference) | 1 (reference) | – | – | – |
| Lymphocyte count, /mm$^3$ | | | | |
| <800 | 2.84 (1.41–5.72) | 0.003 | – | – |
| ≥800 and <1000 | 1.50 (0.66–3.41) | 0.33 | – | – |
| ≥1000 (reference) | 1 (reference) | | – | – |
| Causative NTM species | | | | |
| *M. abscessus* complex | – | – | 4.30 (1.46–12.69) | 0.008 |
| *M. intracellulare* | – | – | 1.36 (0.59–3.13) | 0.48 |
| *M. avium* | – | – | 1 (reference) | – |
| HRCT pattern of NTM disease | | | | |
| Cavitary NB/fibrocavitary form | 2.92 (1.51–5.65) | 0.002 | 4.08 (1.70–9.80) | 0.002 |
| Unclassifiable form | 1.30 (0.44–3.83) | 0.63 | 1.14 (0.31–4.18) | 0.84 |
| Non-cavitary NB form (reference) | 1 (reference) | – | 1 (reference) | – |

* Adjusted HRs (95% CIs) are shown for variables that remained in the final Fine-Gray models.

Fine-Gray competing risks analyses were conducted to evaluate the baseline patient characteristics that predict all-cause mortality and NTM-related mortality over time. All predictor variables with *p*-values <0.1 in Gray's test shown in Table 5 were included in Fine-Gray regression analyses. Abnormal HRCT findings were not included in these analyses together with HRCT patterns of pulmonary NTM disease because both predictor variables were highly correlated. A backward stepwise selection with a cut-off significance level of 0.05 was used as the variable selection procedure in each regression analysis.

RA, rheumatoid arthritis; NTM nontuberculous mycobacteria; NB form, nodular/bronchiectatic form; HRs, hazard ratios; CIs, confidence intervals.

patients with NTM disease who had received anti-TNF therapy between 2000 and 2008, and found that seven patients (39%) died with a median time between infection and death of 569 days [21]. Among the death cases, five continued to receive anti-TNF agents after the diagnosis of NTM disease. It is uncertain whether or not they received anti-NTM therapy concomitantly. In a retrospective chart review of 13 patients who had developed pulmonary NTM disease during bDMARD therapy for RA, Mori et al. showed that, following the discontinuation of bDMARDs, most patients responded to anti-NTM therapy and no deterioration of radiological findings was observed in any patients [28]. In a case series study of 11 patients with RA receiving bDMARDs after diagnosis of pulmonary NTM disease, Yamakawa et al. showed that radiological deterioration was not observed in the majority (64%) of these patients. In some patients undergoing anti-NTM therapy, radiological outcomes of NTM disease were favorable [36]. Several cases of the successful continuation or start of bDMARDs after diagnosis of pulmonary NTM disease have also been reported, in which adequate anti-NTM drugs were concomitantly used during bDMARD therapy [20, 37, 38]. In the present study, the continuation or start of bDMARD or tsDMARD after the diagnosis of NTM disease was not identified as a factor associated with estimates of all-cause or NTM-related mortality over time, which might be explained by the strict instruction related to the concomitant use of anti-NTM drugs.

Although the use of bDMARDs has been recognized as a risk factor for the development of NTM disease, it seems unlikely to increase the risk of death in RA patients concomitantly receiving anti-NTM therapy. The survival benefits of the long-term use of a macrolide as anti-NTM therapy to patients with NTM infection were recently reported [12].

Through a systematic review of patients with pulmonary MAC disease, Diel et al. identified 14 eligible studies from the literature up to August 2017, and showed that the pooled estimate of 5-year all-cause mortality was 0.27 [39]. A high degree of heterogeneity was observed across studies, ranging from 0.1 to 0.48. Predictive factors of all-cause mortality consistent across studies included male sex, presence of comorbidities, and advanced age. In addition, several studies reported a higher risk of death in patients with cavitary NTM disease [29, 30, 40–43]. Hypoalbuminemia and lymphocytopenia were also reported as predictors of overall mortality [41, 43]. In a most recent retrospective study including 1445 patients newly diagnosed with pulmonary NTM disease caused by MAC or *M. abscessus* between 1997 and 2013 at a single referral hospital in South Korea, Jhun et al. reported that the 5-, 10-, and 15-year cumulative all-cause mortality rates were 0.12, 0.24, and 0.36, respectively [44]. Causative NTM species (*M. abscessus*), cavitary disease, and some demographic characteristics such as advanced age and male sex were significantly associated with long-term all-cause mortality. Through an analysis of the nationwide database of South Korean National Health Insurance, Lee et al. showed that the overall 6-, 10-, and 14-years cumulative survival probabilities were 75.1%, 65.4%, and 57.0%, respectively [12]. Advanced age, male gender, provincial area, and comorbidities were significant factors associated with the mortality of NTM infection. In the present study, the cumulative incidence of all-cause death at 5 years was estimated to be 0.22, and similar baseline variables to those studies were identified as the predictive factors for death outcomes.

Several studies showed that respiratory comorbidities, particularly ILD, emphysema, past TB, and chronic obstructive pulmonary disease (COPD), were predictive factors for all-cause death in patients with pulmonary NTM disease, although their effect was less potent compared with advanced age and male gender [12, 30]. Mirsaeidi et al. indicated that compared to TB-related mortality, COPD, bronchiectasis, and ILD were significantly more common in patients with NTM-related death [11]. Diel et al. showed that the mortality rate was significantly higher in COPD patients with pulmonary NTM disease compared with those without NTM disease [45]. For RA patients, Yamakawa et al. indicated that underlying lung disease was present in 50% of patients (16% for ILD, 9% for emphysema, 7% for past TB, and 7% for bronchiolitis). Survival probabilities were significantly different between patients with usual interstitial pneumonia or emphysema and those without underlying lung disease. In a Cox regression analysis, however, these comorbidities were not identified as prognostic factors for all-cause mortality [31]. In the present study, ILD and past TB were factors associated with all-cause death in univariate analyses, but did not remain as predictive factors in multivariate Fine-Gray regression analysis. Although patients with chronic pulmonary disease apparently have an increased risk of developing pulmonary NTM disease [4, 5, 7, 46], patient demographical characteristics, causative NTM species, and the presence of cavitary disease appear to contribute more to mortality in pulmonary NTM patients.

In the present study, the mortality estimates of NTM-related or all-cause death in RA patients were not significantly different between RA patients and non-RA patients, which might be explained by the similarity in baseline patient characteristics (age, comorbidities, and laboratory data) as well as characteristics of NTM disease (NTM species and HRCT patterns) between both patient groups in our cohort. The predominance of female RA patients may have contributed to this result. Although it is well recognized that the risk of the development of pulmonary NTM disease is higher in RA patients compared with non-RA patients [13–15],

pulmonary NTM disease may have similar clinical features, prognostic factors, and outcomes between RA and non-RA patients.

There are several limitations to this study. First, this study was conducted in two community hospitals located in the Kyushu region of Japan. Therefore, our results may not be generalizable to other geographical areas. In addition, our institutions are tertiary referral centers and, therefore, selection bias cannot be entirely excluded. Second, the number of RA patients who had been newly diagnosed with pulmonary NTM disease was small in this study. Therefore, it is unlikely that the study reflects the complete characteristics of pulmonary NTM disease occurring in RA patients. Large-scale population-based registry studies are warranted to confirm our results. Third, since the main aim of the present study was not to evaluate radiological deterioration on serial HRCT scans following the diagnosis of pulmonary NTM disease, we did not calculate the detailed scores for each of the abnormal HRCT findings according to the scoring system for quantification of the extent of NTM disease [47, 48]. Therefore, we refrained from adopting each of these abnormalities for Fine-Gray regression analysis as predictor variables. Fourth, we could not include the rate of completion of anti-NTM therapy in survival analyses, because this variable was determined after starting follow-up (i.e., it was a time-varying variable). However, the similar completion rate observed in this study may suggest that this rate might have little effect on all-cause or NTM-related mortality. Finally, this was a retrospective cohort study, which may confer certain inherent limitations. Some clinical and laboratory findings were not available. Body mass index, which has been reported to be associated with poor prognosis in the general population [30, 41, 43, 44], was not always measured at the time when the definitive diagnosis of pulmonary NTM disease was made.

## Conclusions

According to Fine-Gray competing risks regression analysis, advanced age (≥70 years), male sex, *M. abscessu*s complex as the causative species, and the presence of cavitary disease were the predictive factors for NTM-related death in patients who were newly diagnosed with pulmonary NTM disease. Contrary to our expectations, RA patients with pulmonary NTM disease did not exhibit a greater risk of long-term mortality compared with non-RA patients. When predicting NTM disease outcomes in RA patients, clinicians should instead consider patients' demographic characteristics, causative NTM species, and the presence of cavitary disease, all of which have been generally recognized as the predictive factors for mortality in patients with pulmonary NTM disease.**References**

## Supporting information

**S1 Table. Predictive factors for mortality in patients with pulmonary NTM disease (Fine-Gray models using a forced-entry method).**
(DOC)

## Author Contributions

**Conceptualization:** Shunsuke Mori, Yukitaka Ueki.

**Data curation:** Shunsuke Mori, Yukinori Koga, Kazuyoshi Nakamura, Sayuri Hirooka, Takako Matsuoka, Hideshi Uramoto, Osamu Sakamoto, Yukitaka Ueki.

**Formal analysis:** Shunsuke Mori, Yukinori Koga, Yukitaka Ueki.

**Funding acquisition:** Shunsuke Mori.

**Investigation:** Shunsuke Mori, Yukinori Koga, Kazuyoshi Nakamura, Sayuri Hirooka, Takako Matsuoka, Hideshi Uramoto, Osamu Sakamoto, Yukitaka Ueki.

**Methodology:** Shunsuke Mori, Yukitaka Ueki.

**Project administration:** Shunsuke Mori.

**Resources:** Shunsuke Mori, Kazuyoshi Nakamura, Sayuri Hirooka, Takako Matsuoka, Hideshi Uramoto, Osamu Sakamoto, Yukitaka Ueki.

**Supervision:** Shunsuke Mori, Osamu Sakamoto.

**Validation:** Shunsuke Mori, Yukinori Koga, Kazuyoshi Nakamura, Osamu Sakamoto, Yukitaka Ueki.

**Visualization:** Shunsuke Mori.

**Writing – original draft:** Shunsuke Mori, Yukinori Koga, Kazuyoshi Nakamura, Sayuri Hirooka, Takako Matsuoka, Hideshi Uramoto, Osamu Sakamoto, Yukitaka Ueki.

**Writing – review & editing:** Shunsuke Mori, Yukinori Koga, Kazuyoshi Nakamura, Sayuri Hirooka, Takako Matsuoka, Hideshi Uramoto, Osamu Sakamoto, Yukitaka Ueki.

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
