## [Decision Letter · Decision Letter 0]

2 Oct 2020

PONE-D-20-28997

Mortality in rheumatoid arthritis patients with pulmonary nontuberculous mycobacterial disease: A longitudinal cohort study

PLOS ONE

Dear Dr. Mori,

Thank you for submitting your manuscript to PLOS ONE. After careful consideration, we feel that it has merit but does not fully meet PLOS ONE’s publication criteria as it currently stands. Therefore, we invite you to submit a revised version of the manuscript that addresses the points raised during the review process.

All reviewers found some interests in this study, but pointed out a number of criticisms that require improvement and amendment. I ask the authors to fully respond to all comments made by reviewers in the revised version.

We look forward to receiving your revised manuscript.

Kind regards,

Masataka Kuwana, MD, PhD

Academic Editor

PLOS ONE

Journal Requirements:

Reviewers' comments:

Reviewer's Responses to Questions

**Comments to the Author**

1. Is the manuscript technically sound, and do the data support the conclusions?

Reviewer #1: Yes

Reviewer #2: Yes

Reviewer #3: Yes

2. Has the statistical analysis been performed appropriately and rigorously? 

Reviewer #1: Yes

Reviewer #2: Yes

Reviewer #3: Yes

3. Have the authors made all data underlying the findings in their manuscript fully available?

Reviewer #1: No

Reviewer #2: Yes

Reviewer #3: Yes

4. Is the manuscript presented in an intelligible fashion and written in standard English?

Reviewer #1: Yes

Reviewer #2: Yes

Reviewer #3: Yes

5. Review Comments to the Author

Reviewer #1: Mori et al. examined the mortality in rheumatoid arthritis (RA) patients with pulmonary nontuberculous mycobacterial (NTM) disease by a retrospective study. The authors should elucidate the following issues:

1. The title is not incorrect but misleading, because this is a retrospective study.

2. Non-RA controls must be specified. Did they include patients with other rheumatic diseases such as Sjögren syndrome (Chao et al. BMC Infectious Diseases 2017) and others (Takenaka et al. Mod Rheumatol 2020)? Because this article did not describe laboratory data such as serum albumin level (Kim et al. BMC Pulmonary Medicine 2017) and lymphocyte count (Hirose et al. Mod Rheumatol 2019), the description of controls is crucial.

3. Some descriptions in Statistical analysis section may be moved to elsewhere.

4. Death caused by the exacerbation of pulmonary NTM disease may be demonstrated by the chest images.

5. Figure 1 and 2: The labeling for the y-axis should be improved for the understanding without figure legends.

6. The identification of age ≥80 as a mortality predictor is universal and clinically meaningless. How about ≥70?

Reviewer #2: Mori et al. analyzed newly diagnosed 225 pulmonary NTM cases to clarify the predictive factors of death outcomes. They focused on the differences between cases with/without RA. The methodologies and statistical analyses are suitable, and the manuscript is well written. However, the conclusions were the same as the previous studies, and there are several concerns in this manuscript.

#. I understand that the authors tried to clarify whether RA could influence on the NTM mortality. However, the number of cases in this study is too small to generalize the results. Furthermore, the study was conducted in the tertiary hospitals for NTMs and RA. Although they describe geographical area bias as a limitation, the results cannot be generalized, and the population-based or large scare registry studies are warranted.

#. HRCT findings; they analyzed nodules, tree-in-bud sign, and consolidation, but the results were not shown in the results section.

#. Statistic: I am not sure why they used stepwise selection in this study. Please ask a statistician.

#. Ethical; the study was approved in the Kumamoto Saishun Medical Center, but there was no description of the Sasebo Chuo Hospital's approval. What is the meaning of the following sentence? "this has been approved by our ethical committee."

#. In the previous large scale study conducted in South Korea, M. intracellulare was a risk factor for deaths. So it would be better to divide into M. avium and M. intracellulare.

#. M. abscessus should be "M. abscessus complex."

#. Cavitary opacity is better to be "Cavitary lesion."

#. They need to explain the cumulative incidence function in the method section. Similarly, in Line 266, "all-cause death at five years was 0.24". What the unit of 0.24?

#. The footnote of Table 6, Gray's scale test shown in Table "4", and in the following line, NTT should be "NTM."

#. Figure 1B: The cumulative incidence of death in RA cases seems to be increasing after six years.

Reviewer #3: Comments to the Author

Mori al. report a study on the focusing between RA-NTM and non-RA-NTM. This focus of authors is unique and interesting, though this study does not extremely original results.

★comments

Patients with RA have an increased risk of infection compared with the general population. Therefore, I expected the result that survival for patients RA-NTM would worse than those for patients with non-RA-NTM. Because sample size of this study is relatively small, this might lead to those results. However, it is interesting to me.

Authors mentioned that RA patients with p-NTM disease were not greater risk of long-term mortality compared with non-RA patients. In other words, because there was no significant difference of prognosis for NTM as whether RA or not, radiological cavitary disease, older, and male were thought to be important for prognosis of each patient. This matter should be emphasized for more including author’s speculation.

In addition, I thought that the present study may mean to most important factor of comorbidity for chronic pulmonary disease (i.e., interstitial lung disease, pulmonary emphysema, and old pulmonary tuberculosis). I would like you to discuss in this point.

6. PLOS authors have the option to publish the peer review history of their article (what does this mean?). If published, this will include your full peer review and any attached files.

Reviewer #1: No

Reviewer #2: No

Reviewer #3: No

---

## [Author Response · Author response to Decision Letter 0]

5 Nov 2020

Response to Reviewers

We are most grateful to the reviewers for their valuable comments. We have made all requested changes and added new information to the manuscript in response to their insightful comments. All alterations are highlighted in red text in the revised version of the manuscript. We are confident that the manuscript has benefited from the reviewers’ useful comments and suggestions.

Below are point-by-point replies to the reviewers’ comments.

Reply to Reviewer 1

1. Thank you for this helpful comment. Since longitudinal studies can be retrospective or prospective, we used “a longitudinal cohort study” in the original title. We understand the reviewer’s concern that this title may be misleading, however. As such, in the revised version, we modified the title as follows: “Mortality in rheumatoid arthritis patients with pulmonary nontuberculous mycobacterial disease: A retrospective cohort study” (lines 2–3). We also added the word “retrospective” to several sentences in the text (lines 88 and 350), for clarity 

2. We agree with your comment that we should specialize non-RA controls. Among the non-RA group in this study, only one patient had autoimmune rheumatic disease other than RA (ANCA-associated vasculitis). There were no patients with Sjögren’s syndrome or other rheumatic diseases (systemic lupus erythematosus, dermatomyositis, etc.). This information was added to the Materials and Methods section in the revised version (lines 118–119). In addition, we included baseline serum albumin and lymphocyte count for each patient as predictor variables in the survival analyses for all-cause death and NTM-related death in the revised version. We obtained the following new data: (1) the mean lymphocyte count at baseline was lower in RA patients compared with non-RA patients; (2) in univariate analyses (Gray’s test), hypoalbuminemia (<3.0 g/dl) and lymphocytopenia (<800/mm³) were associated with cumulative incidence probability over time for all-cause death and NTM-related death; (3) according to Fine-Gray regression analyses, hypoalbuminemia and lymphocytopenia were significant predictive factors for all-cause death, not NTM-related death, in pulmonary NTM disease patients. We added new data and information to the text (lines 129–132, 254–255, 315–324, 329–334, 358–360, and 408–409) as well as Tables 1, 5, and 6 of the revised manuscript.

3. We appreciate your suggestion to move some of the descriptions in the Statistical Analysis section. In response to this comment, we removed the descriptions of predictor variables (lines 195–201 of the previous version) and placed some of them in the Results section of the revised version (lines 313–315). All of the predictor variables used in the survival analyses are shown in Table 5. 

4. We appreciate your suggestion that exacerbation of pulmonary NTM disease may be demonstrated by chest images. In response to this suggestion, we included HRCT images of a patient with the cavitary nodular/bronchiectatic (NB) form (case 5 in Table 4) as Fig. 2 of the revised version. HRCT scans were taken at the time of diagnosis of pulmonary NTM disease and 8 months before the patient died. This patient discontinued etanercept monotherapy after NTM disease was suspected and completed anti-NTM therapy. Radiological findings were exacerbated, however, and the patient eventually died. In addition, we included several sentences in the revised Results section (lines 306–307) and added the figure legend for Fig. 2 (lines 682–689). 

5. We appreciate your comment on the labeling for the y-axis of Figs. 1 and 2 in the previous version. Cumulative incidence of death is defined as the probability that a death event occurs before a given time. In the present study, we used the cumulative incidence function (CIF) instead of the Kaplan–Meier (K–M) survival function to estimate the probability of the occurrence of a death event over time, because we considered the presence of competing risks in the survival analyses (lines 206–211). In the K–M survival curves, “survival probability” or “survival percentage” is often used as the label for the y-axis. In the CIF plots, the software for statistical analysis (Easy R), which we used for this study, labeled the y-axis with “cumulative incidence” with a scale of 0.0–1.0. In the present study, we used “cumulative incidence” to clearly show that the analysis has been done with the CIF (not the K–M function). In response to your comment, we added “probability of all-cause death” or “probability of NTM-related death” to the label for the y-axis of Fig. 1 and “probability of death” to Fig. 3 in the revised version. In addition, we modified several sentences in the abstract and text (lines 39–42 and 284–286).

6. We agree with your comment that age ≥80 years as a predictive factor of death is clinically meaningless. In response to this comment, we categorized the patients into three groups (age ≥80, ≥70 and <80, and <70 years), and then performed survival analyses using age <70 years as a reference. According to the Fine-Gray regression analysis, advanced age (≥80 and 70 – <80) was a significant predictive factor for all-cause death (adjusted HR 3.79 [95% CI 1.82–7.89] for ≥80 years and 2.56 [1.27–5.16] for 70 – <80 years vs. <70 years) and NTM-related death adjusted (HR 7.28 [95% CI 2.91–18.20] for ≥80 years and 3.68 [1.46–9.26] for 70 – <80 years vs. <70 years). We included these results in the abstract and text (lines 43–47, 303, 315–324, 329–338, 353–356, 358–360, and 478–481) of the revised version. We also added new data to Tables 1, 5, and 6. Figure 3A (cumulative incidence of NTM-related death grouped according to age) and its figure legend (lines 694–695) were also modified. 

Regarding the data availability, all data underlying the findings in this study are available, without restriction, from the Human Research Ethics Committee of the National Hospital Organization Kumamoto Saishun Medical Center (contact information: Mr. Shunichi Tsutsumiuchi, Control Manager of the Committee, tsutsumiuchi.shunichi.dz@mail.hosp.go.jp) for all interested researchers who meet the criteria for access to confidential data. These data include potentially identifying personal information of individual patients. Data that are not directly identifiable are also inappropriate to share publicly, because, in combination, these data can become identifying, especially data collected from the RA group with NTM disease and the NTM-related death group. Therefore, the Ethics Committee does not recommend that such data be made public unnecessarily. Please understand that these strict rules for protecting participant privacy are imposed on us by the Ethics Committee.

Reply to Reviewer 2

Limitations

We understand your concern about the generalization of this study. As you pointed out, our institutions are tertiary referral centers and, therefore, selection bias cannot be entirely excluded. In addition, the number of RA patients who had been newly diagnosed with pulmonary NTM disease was small in this study. Therefore, it seems unlikely that the study reflects the complete characteristics of pulmonary NTM disease occurring in the RA population. Population-based or large-scale registry studies are warranted. We added these limitations to the Discussion section of the revised version (lines 457–461).

HRCT findings

We appreciate your comment that the results regarding abnormal HRCT findings (tree-in-bud, nodules, and consolidation) were not presented in the previous version. In response to this comment, we added the number of patients with each of these abnormal findings to Table 1 and the Results section (lines 263–265). In addition, we compared the estimated cumulative incidence of death over time between patient groups classified according to each of the abnormal HRCT findings, using the cumulative incidence function (CIF) with Gray’s test. Cavitary lesions and consolidation were variables that were significantly associated with the cumulative incidence of all-cause death. New data were added to Table 5 and the Results section (lines 315–324) of the revised version. 

In the present study, we used these abnormal HRCT findings to determine the HRCT patterns of pulmonary NTM disease for each patient according to the predominant findings and their distribution. Since the main aim of the present study was not to evaluate radiological deterioration on serial HRCT scans following the diagnosis of pulmonary NTM disease, we did not calculate the detailed scores for each of the abnormal HRCT findings according to the previously published scoring system for quantification of the extent and severity of NTM disease (refs. 47and 48). In the revised version, we therefore refrained from adopting each of these abnormalities for Fine-Gray regression analysis as predictor variables. We added this limitation to the Discussion section (lines 461–467) of the revised version. New references (refs. 47 and 48) were also added. Instead of using the abnormal HRCT findings, we used the HRCT pattern as a predictive factor in the revised version. The cavitary nodular bronchiectatic (NB) form and fibrocavitary form were combined and statistically analyzed as the cavitary disease group, because (1) it has been reported that both forms have a similar prognosis (refs. 29–31), (2) we focused on the effect of cavitary lesions on mortality estimates over time; and (3) the number of patients with the fibrocavitary form was small. We modified the Materials and Methods section in the revised version (lines 142–147 and 154–159). New data was also included in the Results section (lines 329–338) as well as Tables 5 and 6 in the revised version.

Statistical analysis

We appreciate your comment on the use of the stepwise selection procedure in the Fine-Gray regression analysis. In the present study, the number of death events were 61 for all-cause death and 42 for NTM-related death. In logistic regression and Cox/Fine-Gray regression analyses, it is generally recommended that one predictor variables be studied for every ten events (the one in ten rule: Peduzzi P, et al. J Clin Epidemiol 1996; 49, 1373-9). For smaller ratios of events per predictor variable, the regression coefficients can be biased in both positive and negative directions. In addition, collinearity can be caused by having too many variables in the same regression analysis. In accordance with the one in ten rule, the number of predictor variables in this study should be limited to 6 for all-cause death and 4 for NTM-related death. In the previous version, to compensate for the small number of death events, we used a stepwise selection as the variable selection procedure in the Fine-Gray regression analyses. To avoid missing predictor variables with clinical relevance and importance, we first screened predictive variables with Gray’s test as univariate analysis. As you pointed out, the selection of predictor variables with p < 0.1 led to fulfilment of the one in ten rule in the previous version, and therefore we did not need to use the stepwise selection procedure. 

During the revision of the manuscript, however, we found additional predictor variables with p < 0.1 in Gray’s test, and therefore we needed to include them in the Fine-Gray analyses’ nine predictor variables (serum albumin, lymphocyte count, past tuberculosis, age, sex, type 2 diabetes, interstitial lung disease, NTM species, and cavitary disease) for all-cause death and 7 (serum albumin, lymphocyte count, past tuberculosis, age, sex, NTM species, and cavitary disease) for NTM-related death. Therefore, we used backward stepwise selection to construct the final Fine-Gray models in the revised version. To address your concern, we newly included results from Fine-Gray regression analyses without stepwise selection (namely, the use of the forced-entry method) as supplementary Table S1 in the revised version. We added new information regarding Fine-Gray regression analysis to the text of the revised version (lines 223–229 and 338–340).

Ethical approval

First comment:

As you pointed out, the approval of the Ethics Committees of the participating institutions is critical, and we are sorry that this information was insufficient in the previous version. The National Hospital Organization (NHO) Kumamoto Saishun Medical Center and Sasebo Chuo Hospital have collaborated in a number of research projects under the approval of both ethics committees. The protocol of this study has been approved by the Human Research Ethics Committees of the NHO Kumamoto Saishun Medical Center (No. 29-45/29-45-2) and the Institutional Review Board of Sasebo Chuo Hospital (No. 2014-12). 

Second comment:

Since the study involved a retrospective review of patient records and the data were analyzed anonymously, our ethical committees waived the requirement of informed consent to participate. 

To clarify these points, we modified several sentences in the Ethical Approval section, and provided the approval number issued by the Institutional Review Board of Sasebo Chuo Hospital (lines 191–197).

M. intracellulare and M. avium

We appreciate your comment on the need to divide the M. avium complex (MAC) group into M. intracellulare and M. avium groups. In response to this comment, we performed all statistical analyses after separating M. intracellulare and M. avium groups. New data and information are now included in Tables 1, 4, 5, and 6 as well as Fig. 3 and its figure legend. We modified the abstract and text (lines 43–47, 255–258, and 334–338).

Terminology of M. abscessus

In response to your comment, we changed “M. abscessus” to “M. abscessus complex” throughout the revised manuscript. 

Cavitary opacity

In response to your comment, we changed “cavitary opacity” to “cavitary lesion” throughout the revised manuscript.

Regarding the CIF

We wish to thank you for the comment regarding the need to explain the CIF. Cumulative incidence of death is defined as the probability that a death event has occurred before a given time. In the present study, we used the CIF to estimate the provability of the occurrence of a death event over time, because we considered the presence of competing risks. The occurrence of a competing risk event precludes the occurrence of the primary event of interest. In the absence of competing risks, the Kaplan–Meier (K–M) survival function can be used to estimate the probability of death. In the presence of competing risks, however, the simple use of the K-M survival function can overestimate the cumulative incidence provability of all-cause and NTM-related death. To avoid this possibility, we used the CIF instead of the K-M survival function. Gray’s test for the CIF model was used to compare estimates of death incidence over time among two or three patient groups. Gray’s test is the analogue to the log-rank test that is used for testing the equality of K–M survival curves between groups. We added these descriptions to the Materials and Methods section (lines 206–220) of the revised version. New references were also added (refs. 33 and 34). 

As mentioned above, the cumulative incidence of death is defined as the probability that a death event has occurred before a given time. In the K–M survival curves, “survival probability” or “survival percentage” is often used as the label for the y-axis. In the CIF plots, the software for statistical analysis (Easy R), which we used for survival analyses in this study, labeled “cumulative incidence” for the y-axis with a scale of 0.0 to 1.0. In the present study, we used “cumulative incidence” to clearly show that the analysis has been done with the CIF (not the K–M function). In response to your comment, we added the following to the Results section of the revised version, “the overall 5-year cumulative death probability was estimated to be 24% for RA patients and 23% for non-RA patients” (lines 284–286). Similarly, we modified the description regarding the cumulative incidence of all-cause death and NTM-related death at 5 years in the Abstract (lines 39–42). We also added the term “probability of all-cause death” or “probability of NTM-related death” to the label of the y-axis of Fig. 1 and “probability of death” to Fig. 3 in the revised version.

Footnote of Table 6

Thank you for pointing out these mistakes. We corrected them in the revised version. 

Fig. 1B

We appreciate your comment that the CIF plots seem different between the RA and non-RA groups after 6 years of follow-up. Gray’s test is a method to compare the cumulative incidence of an event of interest over time, not the cumulative incidence of the event occurring at a specific time-point. Accordingly, the impression obtained from the CIF plots and the result of Gray’s test sometimes differ. This is also observed between the impression of K–M curves and the result of the log-rank test. When comparing K–M curves between groups, we can select the log-rank test, generalized Wilcoxon test, or Tarone-Ware test according to the pattern of survival curves. For CIF plots, Gray’s test is the only reliable method to compare the estimated cumulative incidence probability between patient groups. In the revised version, in addition to Gray’s test, we showed the effect of RA on all-cause of death and NTM-related death over time using univariate Fine-Gray regression analyses. The unadjusted HR (95% CI) of RA versus non-RA was 0.86 (0.38–1.98, p = 0.73) for NTM-related death and 1.34 (0.75–2.40, p = 0.32) for all-cause death. These results were added to the figure legend for Fig. 1 (lines 676–678). 

Reply to Reviewer 3

We are most grateful for your valuable comments and suggestions. As described in the Introduction section, the increased risk of pulmonary NTM disease in RA patients, compared with non-RA patients, has been reported worldwide. In this context, we expected that RA might increase the risk of mortality in patients who had developed pulmonary NTM disease. In the present study, however, we found that the estimated cumulative incidence probability of NTM-related or all-cause death is not greater in the RA group compared with the non-RA group. RA did not contribute to mortality in our patient cohort. Rather, the generally recognized predictive factors, such as advanced age, male sex, NTM species, and cavitary disease, did, which is the point we would like to emphasize in the present study. These opinions are presented in the Abstract and Conclusion sections of the revised manuscript (lines 48–51 and 481–487). 

In the revised version, we discussed a possible reason why the mortality estimates of NTM-related or all-cause death RA patients were not significantly different between RA patients and non-RA patients in this study. This may be explained by the similarity in baseline patient characteristics (age, respiratory or non-respiratory comorbidities, and laboratory data) as well as the characteristics of NTM disease (NTM species and HRCT patterns) between both patient groups in our cohort. The predominance of female RA patients may also have contributed to this result. In addition, the continuation and restart of biological or targeted synthetic antirheumatic drugs appears unlikely to increase the risk of death in RA patients concomitantly receiving anti-NTM therapy. The survival benefits of the long-term use of a macrolide as ani-NTM therapy in patients with NTM infection was reported (ref. 12). Since the number of RA patients who had been newly diagnosed with pulmonary NTM disease was small in this study, however, it is unlikely that the study reflects the complete characteristics of pulmonary NTM disease occurring in RA patients. Large-scale population-based registry studies are warranted to confirm our results. We added this discussion to the revised version (lines 397–401, 444–453, and 457–461). A new reference (ref. 12) was also included in the revised version. 

In the revised version, we evaluated the effect of interstitial lung disease (ILD) and past tuberculosis (TB) on the cumulative incidence probabilities of all-cause and NTM-related death. We found that these comorbidities were associated with all-cause death as evidenced by univariate analyses (Table 5), but did not remain as predictive factors in multivariate Fine-Gray regression analysis (Table 6). Several studies showed that respiratory comorbidities, particularly ILD, emphysema, past TB, and chronic obstructive pulmonary disease (COPD), were predictive factors for all-cause death in patients with pulmonary NTM disease, although their effect was less potent compared with advanced age and male gender (refs. 12 and 30). Mirsaeidi et al. showed that compared to TB-related mortality, COPD, bronchiectasis, and ILD were significantly more common in patients with NTM-related death (ref. 11). Diel et al. showed that the mortality rate was significantly higher in COPD patients with pulmonary NTM disease compared with those without NTM disease (ref. 45). For RA patients, Yamakawa et al. indicated that underlying lung disease was present in 50% of patients (ILD, emphysema, past TB, and bronchiolitis). Survival probabilities were significantly different between patients with usual interstitial pneumonia or emphysema and those without underlying lung disease. In a Cox regression analysis, however, these comorbidities were not identified as prognostic factors for all-cause mortality (ref. 31). Although patients with chronic pulmonary disease are apparently exposed to an increased risk of developing pulmonary NTM disease (refs. 4, 5, 7, and 46), patient demographical characteristics, causative NTM species, and the presence of cavitary disease appeared to contribute more to mortality in pulmonary NTM patients. We included this discussion (lines 424–443) and new references (refs. 7, 12, 45, and 46) in the revised version. 

During the manuscript revision, a population-based study addressing the mortality and prognostic factors in NTM infection was reported from South Korea (ref. 12). We included the results from this study in the text of the revised version (lines 66-68, 400–401, 416–420, and 424–427).

---

## [Decision Letter · Decision Letter 1]

16 Nov 2020

Mortality in rheumatoid arthritis patients with pulmonary nontuberculous mycobacterial disease: A retrospective cohort study

PONE-D-20-28997R1

Dear Dr. Mori,

We’re pleased to inform you that your manuscript has been judged scientifically suitable for publication and will be formally accepted for publication once it meets all outstanding technical requirements.

Kind regards,

Masataka Kuwana, MD, PhD

Academic Editor

PLOS ONE

Additional Editor Comments (optional):

Reviewers' comments:

Reviewer's Responses to Questions

**Comments to the Author**

1. If the authors have adequately addressed your comments raised in a previous round of review and you feel that this manuscript is now acceptable for publication, you may indicate that here to bypass the “Comments to the Author” section, enter your conflict of interest statement in the “Confidential to Editor” section, and submit your "Accept" recommendation.

Reviewer #1: All comments have been addressed

Reviewer #2: All comments have been addressed

Reviewer #3: All comments have been addressed

2. Is the manuscript technically sound, and do the data support the conclusions?

Reviewer #1: Yes

Reviewer #2: Yes

Reviewer #3: Yes

3. Has the statistical analysis been performed appropriately and rigorously? 

Reviewer #1: Yes

Reviewer #2: Yes

Reviewer #3: Yes

4. Have the authors made all data underlying the findings in their manuscript fully available?

Reviewer #1: No

Reviewer #2: No

Reviewer #3: Yes

5. Is the manuscript presented in an intelligible fashion and written in standard English?

Reviewer #1: Yes

Reviewer #2: Yes

Reviewer #3: Yes

6. Review Comments to the Author

Reviewer #1: The manuscript has been intensively and adequately revised according to the reviewer's comments, and now it is suitable for publication.

Reviewer #2: The authors address all of the reviewers' comments and the manuscript has been much improved from the original version. I think the study is worth reporting.

Reviewer #3: Mori et al. analyzed pNTM cases to clarify the predictors of outcomes, by focusing between cases with/without RA. This revised manuscript is well written, however, this conclusion was extremely original.

In large-scale study, Hayashi et al. reported that a multivariate Cox proportional hazard model showed male sex, older age, presence of systemic and/or respiratory comorbidity, non-NB radiographic features, body mass index (BMI) less than 18.5 kg/m2, anemia, hypoalbuminemia, and erythrocyte sedimentation rate greater than or equal to 50 mm/h to be negative prognostic factors for all-cause mortality, and FC or FC1NB radiographic features, BMI less than 18.5 kg/m2, anemia, and C-reactive protein greater than or equal to 1.0 mg/dl to be negative prognostic factors for MAC specific mortality. This study included 36 patients with collagen vascular disease of 634 MAC patients. The present study by Mori et al. included 34 RA patients and 191 non-RA patients. Therefore, it would be natural of similar results between this and a previous study. Despite this limitation, I think there is a value in this study because of limited information in RA-NTM.

7. PLOS authors have the option to publish the peer review history of their article (what does this mean?). If published, this will include your full peer review and any attached files.

Reviewer #1: No

Reviewer #2: No

Reviewer #3: No

---

## [Editor Report · Acceptance letter]

18 Nov 2020

PONE-D-20-28997R1 

Mortality in rheumatoid arthritis patients with pulmonary nontuberculous mycobacterial disease: A retrospective cohort study 

Dear Dr. Mori:

I'm pleased to inform you that your manuscript has been deemed suitable for publication in PLOS ONE. Congratulations! Your manuscript is now with our production department. 

Kind regards, 

on behalf of

Prof. Masataka Kuwana 

Academic Editor

PLOS ONE